# Designer exosomes produced by implanted cells intracerebrally deliver therapeutic cargo for Parkinson's disease treatment

Ryosuke Kojima [1,2], Daniel Bojar [1], Giorgio Rizzi[3], Ghislaine Charpin-El Hamri[4], Marie Daoud El-Baba[4], Pratik Saxena[1], Simon Ausländer[1], Kelly R. Tan[3] & Martin Fussenegger [1,5]

Exosomes are cell-derived nanovesicles (50–150 nm), which mediate intercellular communication, and are candidate therapeutic agents. However, inefficiency of exosomal message transfer, such as mRNA, and lack of methods to create designer exosomes have hampered their development into therapeutic interventions. Here, we report a set of EXOsomal transfer into cells (EXOtic) devices that enable efficient, customizable production of designer exosomes in engineered mammalian cells. These genetically encoded devices in exosome producer cells enhance exosome production, specific mRNA packaging, and delivery of the mRNA into the cytosol of target cells, enabling efficient cell-to-cell communication without the need to concentrate exosomes. Further, engineered producer cells implanted in living mice could consistently deliver cargo mRNA to the brain. Therapeutic catalase mRNA delivery by designer exosomes attenuated neurotoxicity and neuroinflammation in in vitro and in vivo models of Parkinson's disease, indicating the potential usefulness of the EXOtic devices for RNA delivery-based therapeutic applications.

[1] ETH Zürich, Department of Biosystems Science and Engineering, Mattenstrasse 26, 4058 Basel, Switzerland. [2] Graduate School of Medicine, The University of Tokyo, 7-3-1 Hongo, Bunkyo-ku, Tokyo 113-0033, Japan. [3] Biozentrum, University of Basel, Klingelbergstrasse 50/70, 4056 Basel, Switzerland. [4] Département Génie Biologique, Institut Universitaire de Technologie (IUTA), F-69622 Villeurbanne Cedex, France. [5] Faculty of Life Science, University of Basel, Mattenstrasse 26, 4058 Basel, Switzerland. These authors contributed equally: Ryosuke Kojima and Daniel Bojar. Correspondence and requests for materials should be addressed to M.F. (email: martin.fussenegger@bsse.ethz.ch)

Exosomes are currently viewed as specifically secreted vesicles for intercellular communication, and are believed to be involved in various biological processes[1–3]. However, the efficiency of exosomal message (such as mRNA, miRNA, and protein contained in exosomes) transfer is poor, and this has hampered elucidation of their precise roles. On the other hand, they are considered to have potential as RNA drug carriers, based on their biocompatibility, bioavailability, and ability to cross the blood-brain barrier[4–6]. Considering the recent developments of engineered mammalian cell-based theranostic agents, which can be implanted into patients and secrete therapeutic proteins on demand[7, 8], we anticipated that mammalian cells capable of being implanted in patients and secreting therapeutic exosomes loaded with biopharmaceutical-encoding mRNAs in-situ would also have potential therapeutic applications[8, 9]. However, the capability to create designer exosomes is still lacking, and current strategies to use exosomes as therapeutic agents still require ex vivo concentration of exosomes and RNA electroporation, due to the inefficiencies of production and message transfer. To overcome these challenges, a new design strategy for creating designer exosomes with dramatically increased and controllable efficiency of exosomal communication has been needed. For this purpose, we focused on engineering the following processes: (1) exosome biogenesis, (2) packaging of specific RNAs into exosomes, (3) secretion of exosomes, (4) targeting, and (5) delivery of mRNA into the cytosol of target cells.

Here, we report a series of synthetic biology-inspired control devices that we call EXOsomal Transfer Into Cells (EXOtic) devices, which serve to enhance these steps, enabling efficient exosomal mRNA delivery without the need to concentrate exosomes. We confirm the functionality of the engineered exosome producer cells in vivo. Moreover, we demonstrate the potential therapeutic usefulness of designer exosomes produced by cells engineered with EXOtic devices by developing an EXOtic therapy that attenuated neurotoxicity and neuroinflammation in in vitro and in vivo models of Parkinson's disease.

## Results

**Development of an exosome production booster**. To boost exosome production by increasing exosome biogenesis and secretion, we first conducted a screen in HEK-293T cells to find genes that enhance exosome production. For this purpose, we prepared a reporter construct by fusing nanoluc (nluc), a small and potent bioluminescence reporter[10], to the C-terminus of CD63 which is one of the most widely used exosome markers[11]. This reporter gene was co-transfected with plasmids encoding candidates for exosome production enhancement, and luminescence in the cell-culture supernatant was measured after stepwise centrifugation to remove masking signals[12] (Fig. 1a, b, Supplementary Fig. 1). We identified STEAP3 (involved in exosome biogenesis[13–15]), syndecan-4 (SDC4; supports budding of endosomal membranes to form multivesicular bodies[16, 17]), and a fragment of L-aspartate oxidase (NadB; possibly boosts cellular metabolism by tuning up the citric acid cycle[18]) as potential synthetic exosome production boosters. Combined expression of these genes significantly increased exosome production, and a tricistronic plasmid vector (pDB60, hereinafter referred to as exosome production booster), which ensures that transfected cells receive all boosted genes at a fixed ratio[19, 20], produced a 15-fold to 40-fold increase (depending on cell conditions) in the luminescence signal in the supernatant (Fig. 1b). Most of the luminescence signal was derived from vesicle-associated CD63-nluc, and not from soluble nluc (Supplementary Fig. 2a, b). We also confirmed the effect of the booster by using another reporter, CD9-nluc, indicating the efficacy of the booster for different

subpopulations of exosomes (Supplementary Fig. 2c). Additionally, we confirmed the boosting of exosome production by direct quantification of exosomal proteins CD9 and TSG101 (Supplementary Fig. 2d–f). Most importantly, nanoparticle-tracking analysis (NTA) showed a dramatic fold increase in production of exosomes without changing their size distribution (Fig. 1c, Supplementary Fig. 3). Altogether, these results provide strong evidence that exosome production was indeed enhanced. It is noteworthy that this exosome production booster was functional in several other cell lines as well (Supplementary Fig. 4), including patient-derived human mesenchymal stem cells (hMSCs) (Fig. 1d), indicating the generalizability of the device.

**Delivery of exosomal mRNA with EXOtic devices**. Next, we set out to develop an active packaging device of specific RNAs into exosomes (hereinafter referred to as RNA packaging device) and a device to help to deliver RNA into the cytosol of target cells (hereinafter referred to as cytosolic delivery helper) (Fig. 2a). For the RNA packaging device, we focused on the archaeal ribosomal protein L7Ae which binds to the C/D_box RNA structure[21–23]. We conjugated L7Ae to the C-terminus of CD63, and inserted a C/D_box into the 3′-untranslated region (3′-UTR) of the reporter gene, hypothesizing that transcripts encoding the reporter protein nluc could be well incorporated into exosomes via the interaction between L7Ae and the C/D_box in the 3′-UTR (Fig. 2a). As a potential cytosolic delivery helper, we focused on a gap junction protein, connexin 43 (Cx43), which was recently reported to be enriched in exosomes and to enhance information transfer from exosomes to target cells through the formation of hexameric channels[24], and its constitutively active mutant S368A. We co-transfected the potential RNA packaging device and cytosolic delivery helper together with the exosome production booster and a reporter coding for nluc mRNA bearing C/D_box(es) in its 3′-UTR, as well as a targeting module, RVG-Lamp2b, which was reported to target exosomes to the brain by binding to nicotinic acetylcholine receptor (CHRNA7)[25, 26]. The supernatant containing the designer exosomes was applied to target HEK-293T cells expressing CHRNA7 without further concentration of the exosomes, followed by a luminescence assay after translation from nluc mRNA delivered to the target cells. As a result, strong luminescence from target cells was detected only when all the components were present (Fig. 2b), indicating the functionality of each device. Depletion of the exosomes from the culture supernatant of exosome-producing cells substantially decreased luminescence, indicating that the signal was indeed generated by transfer of exosomes (Supplementary Fig. 5). The number of C/D_box repeats significantly affected the efficiency of communication (Fig. 2c) (especially as almost no luminescence was detected without C/D_box). Also, the constitutively active Cx43 S368A mutant was a more efficient cytosolic delivery helper than wild-type Cx43 (Supplementary Fig. 6). Thus, we have identified and developed the components required for efficient exosomal cell-to-cell communication and designated them as EXOtic devices. The whole set of EXOtic devices was also functional in patient-derived hMSCs (Fig. 2d). We used HEK-293T cells as exosome producer cells for all follow-up experiments because of their ease of handling and transfection, as well as their higher exosome-production capacity.

**Exosomal catalase mRNA delivery rescues neurotoxicity**. Next, we assessed the ability of the designer exosomes to deliver therapeutically relevant mRNA into target cells. For this purpose, we focused on the treatment of Parkinson's disease, which is a degenerative disorder of the central nervous system[27]. Neuronal cell death is an important aspect of Parkinson's disease and the

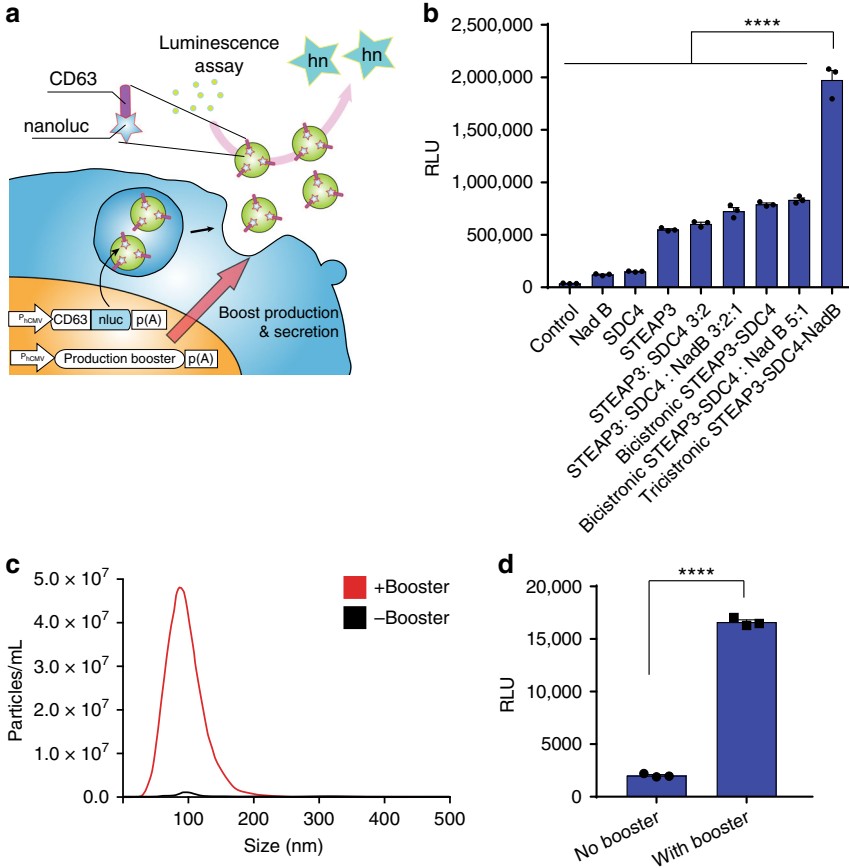

**Fig. 1** Devices to boost exosome production. **a** Schematic illustration of luminescence assay for the quantification of exosome production. Ectopically expressed CD63-nluc is packaged into exosomes, which are secreted into the supernatant by the cultured cells. The ability of transgenes to increase the production of exosomes is determined by measuring the luminescent reporter in the supernatant after stepwise centrifugation to remove live cells, dead cells, and cell debris. **b** Effect of exosome production boosters in HEK-293T cells. CD63-nluc (pDB30: $P_{hCMV}$-CD63-nluc-pA) was expressed together with potential production boosters as follows. Control; EYFP-C1 ($P_{hCMV}$-EYFP-pA), NadB; pRK246 ($P_{hCMV}$-NadB fragment-pA) SDC4; pDB13 ($P_{hCMV}$-SDC4-pA) STEAP3; pDB12 ($P_{hCMV}$-STEAP3-pA), STEAP3:SDC4 3:2; cotransfection of pDB12 and pDB13 (ratio 3:2), STEAP3:SDC4:NadB 3:2:1; cotransfection of pDB12, pDB013, and pRK 246 (ratio 3:2:1), Bicistronic STEAP3 -SDC4; pDB59 ($P_{hCMV}$-STEAP3-IRES-SDC4-pA), bicistronic STEAP3-SDC4: Nad B 5:1; cotransfection of pDB59 and pRK246 (ratio 5:1), tricistronic STEAP3-SDC4-NadB; pDB60 ($P_{hCMV}$-STEAP3-IRES-SDC4-IRES-NadB-pA). Measured values are reported in relative luminescence units (RLU). **c** Result of nanoparticle tracking analysis. Concentration and size distribution of exosomes secreted from cells engineered with the exosome production booster (pDB60) were compared with those under the control (EYFP expression by EYFP-C1) conditions. Raw data is shown in Figure S2. **d** Exosome production booster in patient-derived mesenchymal stem cells (hMSCs). CD63-nluc (pDB30) and the exosome production booster (pDB60) were introduced into hMSCs by electroporation and CD63-nluc secreted into the supernatant was assayed. All the data are mean ± SEM of three independent experiments ($n = 3$). ****$p < 0.0001$, two-tailed Student's $t$-test

delivery of catalase is known to attenuate this cell death by protecting neurons from oxidative damage[28]. Since exosomes can cross the blood-brain barrier[4–6, 29], we assessed whether our designer exosomes bearing catalase mRNA produced by exosome producer cells equipped with the EXOtic devices could rescue neuronal cell death induced by 6-hydroxydopamine (6-OHDA). 6-OHDA is widely used to trigger experimental Parkinson's disease, as it damages neurons in part, though not exclusively, by producing cytotoxic levels of reactive oxygen species (ROS)[30, 31]. Indeed, designer exosomes produced by the engineered exosome producer cells significantly reduced the neurotoxicity of 6-OHDA towards CHRNA7-positive Neuro2A cells without the need for exosome concentration (Fig. 3b), directly indicating that our designer exosomes can deliver biologically functional mRNA with therapeutic potential. Since the present treatment of the 6-OHDA-based Parkinson's model is entirely based on catalase delivery targeting ROS, we see only partial recovery, as ROS-independent cytotoxicities cannot be addressed by catalase treatment[31]. Using a second neurotoxicity model based on

addition of LPS to neuronal and microglial cell co-cultures, we confirmed the rescue of neurotoxicity by designer exosomes containing the catalase mRNA cargo (Supplementary Fig. 7).

**Intracerebral delivery of exosomal mRNA from implant in mice.** Further, we assessed the potential of engineered exosome producer cells (from which exosomes are targeted to the brain by RVG-Lamp2b) in vivo by implanting them in living mice (Fig. 3c). As HEK-293T cells are currently the preferred parental cell line for designer cell-based proof-of-concept studies in vivo[32–35], we continued to use this cell line for following in vivo experiments as well. First, we subcutaneously implanted cells producing CD63-nluc-labelled exosomes and confirmed the presence of the designer exosomes in the bloodstream of treated animals (Supplementary Fig. 8). Second, designer exosomes with nluc mRNA cargo could deliver functional mRNA into brain tissue, inducing expression of the mRNA-coded nluc protein in the target cells (Fig. 3d). The fact that substantially higher nluc

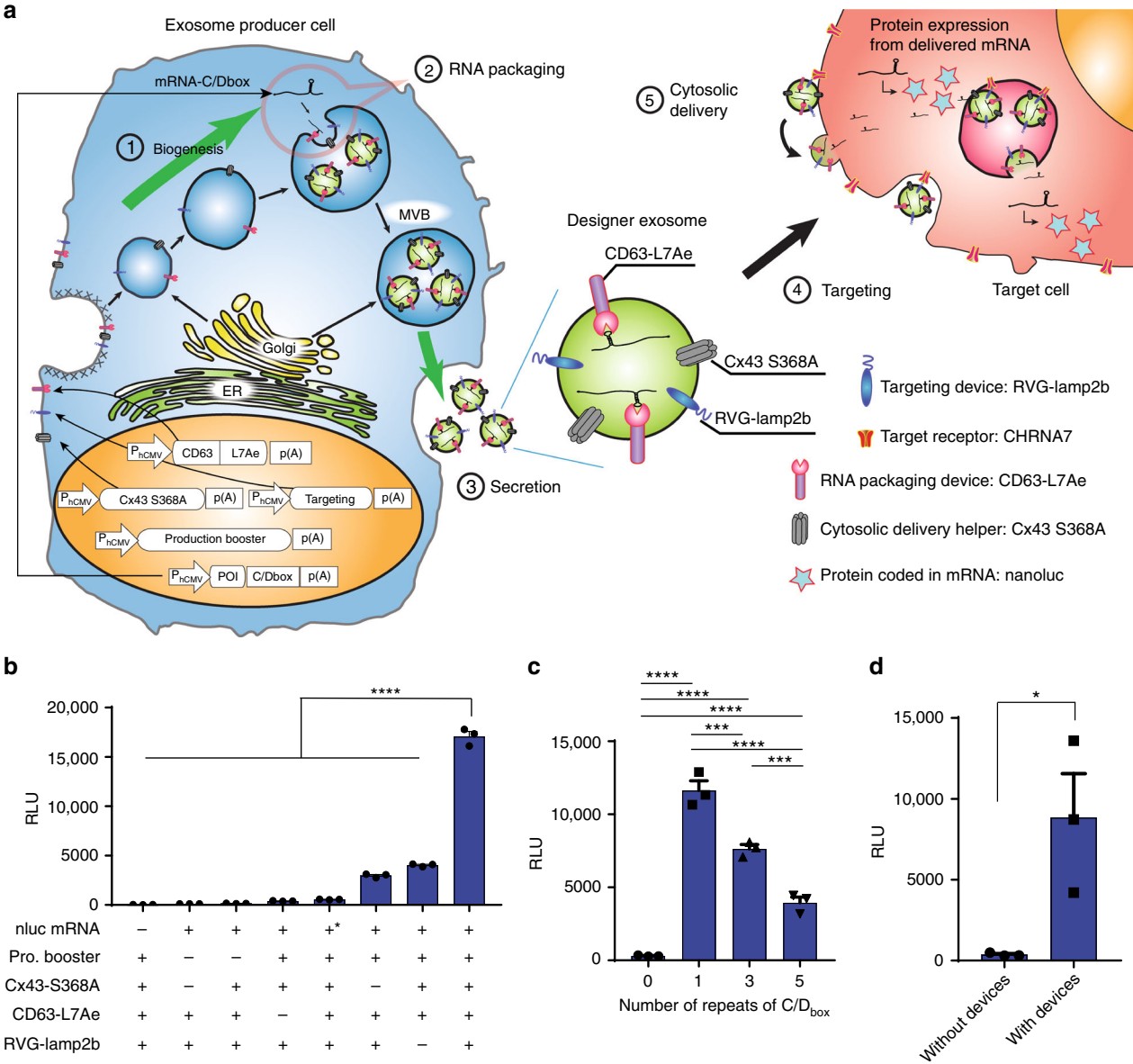

**Fig. 2** EXOtic devices for mRNA delivery. **a** Schematic illustration of the EXOtic devices. Exosomes containing the RNA packaging device (CD63-L7Ae), targeting module (RVG-Lamp2b to target CHRNA7), cytosolic delivery helper (Cx43 S368A) and mRNA (e.g., nluc-C/D$_{box}$) were efficiently produced from exosome producer cells by the exosome production booster. The engineered exosomes were delivered to target cells (HEK-293T cells expressing CHRNA7) and the mRNA was delivered into the target cell cytosol with the help of the cytosolic delivery helper. Finally, protein encoded in the mRNA (e.g., nluc, represented by stars) was expressed in the target cells. **b** Evaluation of the effect of each EXOtic device. HEK-293T cells were transfected with plasmids coding for nluc mRNA (with C/D$_{box}$: pSA462 (P$_{hCMV}$-nluc-C/D$_{box}$-pA), without C/D$_{box}$: pRK0 (P$_{hCMV}$-nluc-pA)), the exosome production booster (Pro. Booster, pDB60), cytosolic delivery helper (pDB68 (P$_{hCMV}$-Cx43 S368A-pA)), RNA packaging device (pSA465 (P$_{hCMV}$-CD63-L7Ae-pA)), and targeting module (pRVG-Lamp2b: P$_{hCMV}$-RVG-Lamp2b-pA[26]) (for (–) condition, pEYFP-C1 was used as compensation). Cell culture supernatant containing engineered exosomes was applied to target cells. The target cell pellet was assayed for nluc activity at 24 h after supernatant transfer. The asterisk indicates nluc mRNA without C/D$_{box}$. **c** Assay of the effect of incorporation of C/D$_{box}$ in the mRNA 3'UTR. HEK-293T cells were transfected with a plasmid coding for nluc bearing 0, 1, 3, or 5 repeats of C/D$_{box}$ in its 3'-UTR (coded by pRK0, pSA462, pSA463, and pSA464, respectively; P$_{hCMV}$-CD63-L7Ae-0~5 of C/D$_{box(es)}$-pA). The cell culture supernatant containing engineered exosomes was applied to target cells and nluc expression was assayed at 24 h after the medium transfer. **d** Evaluation of the whole EXOtic devices in patient-derived hMSCs. Patient-derived hMSCs were transfected with the plasmid coding for nluc-C/D$_{box}$ (pSA462) as well as the EXOtic devices (pSA465, pDB60, pDB68, and pRVG-Lamp2b) or pEYFP-C1 (without device) by electroporation. The cell culture supernatant containing engineered exosomes was applied to target cells and nluc expression was assayed at 24 h after medium application (Note that *Y*-axis of **c** and **d** is not comparable because of several factors including different sender/receiver ratio). Error bars represent SEM of three independent experiments (*n* = 3). *$p < 0.05$, **$p < 0.01$, ***$p < 0.001$, ****$p < 0.0001$, two-tailed Student's *t*-test

activity was observed from the brain in the "with device" condition indicated that the EXOtic devices were functional in vivo and that nluc delivery was indeed mediated by exosomal mRNA transfer. To confirm that successful treatment was based on exosome transfer to the brain, not escape of designer cells from the Matrigel, we macroencapsulated[36] the exosome-producing cells in a device that retained exosome-producing cells but remained permeable to exosomes. As expected, overall delivery efficacy was similar, confirming that the efficacy was due to transfer of the designer exosomes to the brain (Supplementary

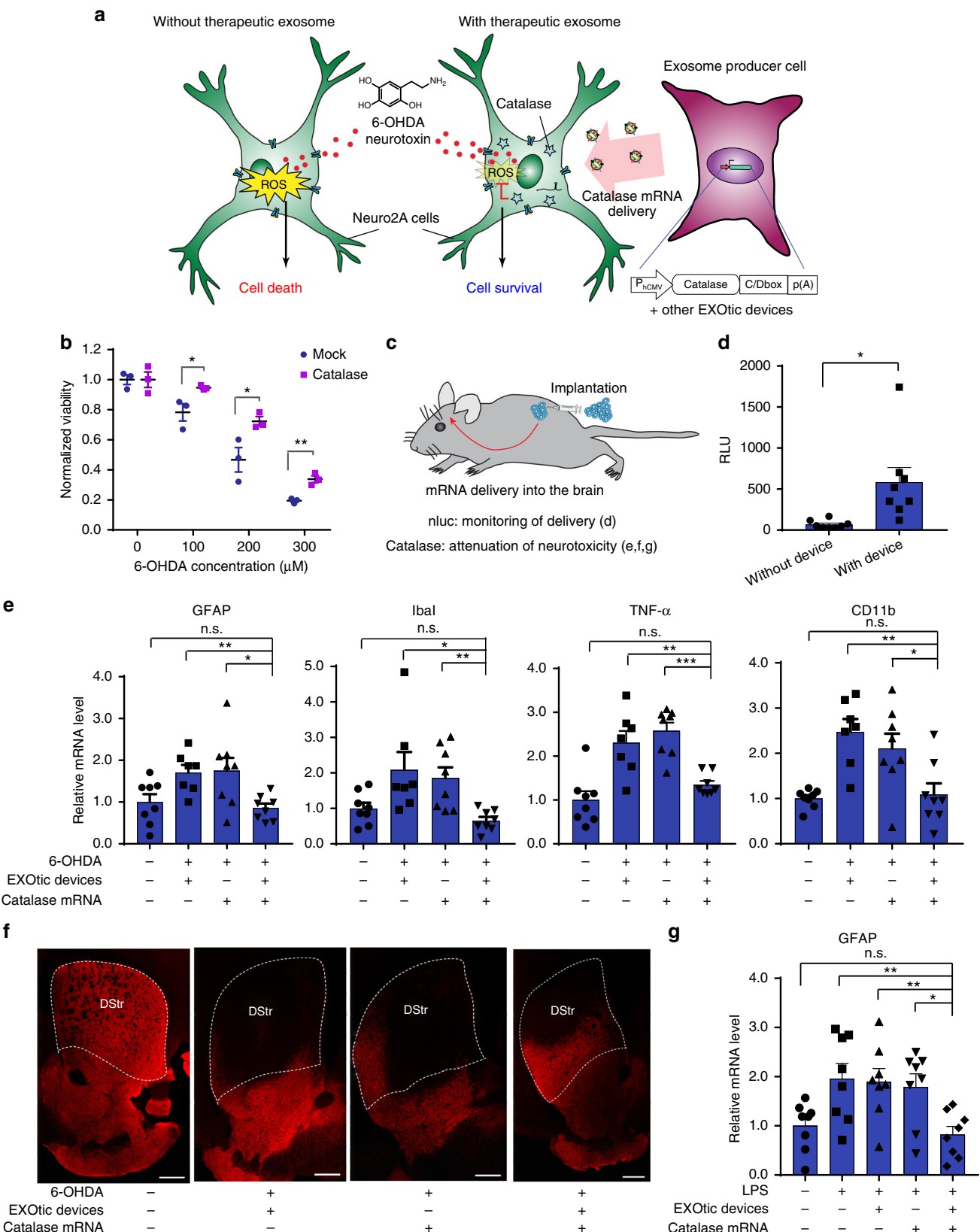

Fig. 9). Biodistribution analysis of exosomes confirmed previous reports that exosomes in the peripheral circulation are not exclusively found in the brain, but also accumulate in spleen and liver[37], which clear most nanoparticulates[38] (Supplementary Fig. 10).

**EXOtic therapy using catalase mRNA for Parkinson's disease.** Next, we tried to deliver therapeutic mRNA into the brain from engineered exosome producer cells implanted in living mice. Neuroinflammation is an important aspect of Parkinson's disease[39, 40]. Since catalase is known to attenuate neuroinflammation[41, 42] resulting from reactive oxygen species produced by neurotoxic reagents such as 6-OHDA, we assessed whether catalase mRNA delivered by the designer exosomes secreted from exosome producer cells implanted in situ in living mice could attenuate neuroinflammation. Following injection of 6-OHDA into the brain of mice we confirmed the attenuation of ROS-triggered neuroinflammation by designer exosomes containing the catalase cargo by profiling the expression levels of several markers (GFAP[28, 43–45], Iba1[45, 46], TNFα[44, 47], CD11b[28]) associated with neuroinflammation in the brain (Fig. 3e). Attenuation of neuroinflammation was not observed following implantation of cells lacking either EXOtic devices or the catalase mRNA expression cassette, indicating that attenuation of neuroinflammation was indeed mediated by designer exosomes containing the catalase mRNA cargo. Although the absolute fold changes of the neuroinflammation markers appear to be moderate, it should be noted that we extracted the mRNA from the whole brain, whereas 6-OHDA injection usually causes only localized neuroinflammation. Importantly, we also showed area-specific rescue of neuronal cell death at the 6-OHDA injection site of the striatum (Fig. 3f).

Since systemic inflammation, which is a known risk factor of Parkinson's disease,[48] causes neuroinflammation, we also confirmed that the therapeutic designer exosomes rescued neuroinflammation induced by systemic injection of LPS (Fig. 3g). Overall, these results confirm that cargo-containing designer exosomes produced by engineered cells implanted in the periphery can deliver therapeutic cargo into the brain to treat experimental Parkinson's disease.

## Discussion

Inspired by synthetic biology, we have successfully developed EXOtic devices enabling efficient customizable production of designer exosomes. Moreover, we showed a potential application of the exosome producer cells equipped with the EXOtic devices

in the treatment of Parkinson's disease by the delivery of catalase mRNA into the brain.

First, we developed an exosome production booster that can be genetically encoded in the exosome producer cells. Then we also succeeded in developing a specific mRNA packaging device as well as a cytosolic delivery helper. Combined use of these devices significantly increased the delivery of exosomal mRNA into target cells. Deleting any component significantly decreased luminescence, indicating that all components are indispensable for efficient transfer. The fact that the number of C/D$_{box}$ repeats affected the efficiency of communication (Fig. 2c) indicated that the balance of association (for RNA packaging) and dissociation (for functionality in target cells) of mRNA and RNA-binding proteins is important. The fact that we could not detect strong luminescence without a C/D$_{box}$ in the 3′-UTR also indicates that the nluc delivery is indeed mediated by mRNA transfer into the exosomes and then the cytosol of target cells. With the constitutively active Cx43 S368A mutant being a more efficient cytosolic delivery helper than wild-type Cx43, the importance of this protein's activity was shown (Supplementary Fig. 6). It is noteworthy that the whole set of EXOtic devices was functional in patient-derived human mesenchymal stem cells (hMSCs), since hMSCs are used as a source for therapeutic exosomes[49] (Figs. 1d, 2d). Thus, the EXOtic devices might serve as important tools to efficiently produce designer exosomes for therapeutic purposes.

Synthetic biology-inspired cell-based treatment strategies are based on implanted designer cells that produce and secrete therapeutic molecules in situ[7, 8, 50]. However, the effector output has so far been limited to the production of secreted protein therapeutics. With the design of the EXOtic devices, we have successfully extended the concept of implanted drug-producing designer cells to in-situ production and delivery of mRNA-containing exosomes. This is particularly important as the constitutive production and delivery of drug-encoding mRNAs inside exosomes may overcome the challenges associated with the limited half-life of exosomes administered intravenously[51]. Indeed, we have shown the potential of the system for therapeutic applications by demonstrating that catalase mRNA delivery can attenuate neurotoxicity and neuroinflammation in both an in vitro and an in vivo model of Parkinson's disease. Although some researchers have shown that delivery of catalase into the brain is sufficient to attenuate neuroinflammation[28], repeated administrations of concentrated exosomes loaded with catalase ex vivo were required to achieve this effect. Implantable exosome producer cells developed in this study allowed for the efficient production and delivery of exosomes containing catalase mRNA in situ for the first time, which could open the door for new ways

**Fig. 3** Application of the EXOtic devices. **a** Protection against neurotoxicity in an in vitro experimental model of Parkinson's disease by catalase mRNA delivery. **b** HEK-293T cells were transfected with plasmids encoding catalase mRNA with a C/D$_{box}$ (pDB129: P$_{hCMV}$-Catalase-C/D$_{box}$-pA) together with the EXOtic devices (pSA465, pDB60, pDB68, and pRVG-Lamp2b) (mock: transfection with pEYFP-C1). Supernatant containing the engineered exosomes was applied to Neuro2A cells (CHRNA7-positive). Then, the cells were incubated with various 6-OHDA concentrations, and CCK-8 viability assay was conducted. **c** Engineered exosome producer cells were subcutaneously implanted with Matrigel in living mice (C57BL6/J) and the mice were used for further assays. **d** Result of nluc mRNA delivery. The exosome producer cells were transfected with pSA462 (P$_{hCMV}$-nluc-C/D$_{box}$-pA) and the EXOtic devices (with device: pSA465, pDB60, pDB68, and pRVG-Lamp2b, without device: pEYFP-C1) and the cells were subcutaneously implanted in mice. Forty-eight hours later, the brains were removed. Luminescence from whole brain homogenates was quantified. **e** Attenuation of neuroinflammation by catalase mRNA delivery in vivo. Exosome producer HEK-293T cells (transfected with pDB129 (P$_{hCMV}$-Catalase-C/D$_{box}$-pA), pSA465, pDB60, pDB68, and pRVG-Lamp2b. pEYFP-C1 was used as compensation for (−) condition) were subcutaneously implanted in mice. One day later, the mice were intracerebrally injected with 2.5 μg of 6-OHDA. Six days later, total RNA was extracted from the whole brain, and mRNA expression level of each marker was assayed by qPCR (GAPDH for internal standard). **f** Immunostaining result of TH-positive neurons. The mice were treated the same way as for **e**. After sacrificing the mice, midbrain sections were prepared, and immunostaining of TH-positive neurons in dorsal striatum (DStr, indicated by a dotted line) was conducted. Ruler bars represent 500 μm. **g** Attenuation of neuroinflammation caused by systemic LPS injection with catalase mRNA delivery in vivo. The same exosome producer cells were prepared as for **e**. At 2 days after implantation, 0 or 0.3 mg/kg of LPS were injected i.p.. 4 h later, RNA was extracted from the whole brain, and GFAP mRNA expression level was assayed by qPCR (GAPDH for internal standard). Error bars represent SEM (**b** three independent experiments (n = 3), **e**, **f**: 7–8 mice (n = 7–8).) *p < 0.05, **p < 0.01, ***p < 0.001, two-tailed student's t-test

to achieve therapeutic effects. Unlike the treatment of metabolic disorders such as diabetes mellitus, which requires daily changing dosing regimens, the successful therapy of Parkinson's disease necessitates continuous high-level drug doses to attenuate any associated neuroinflammation. We have therefore engineered the implanted designer cells for maximum constitutive in-situ production of drug-containing designer exosomes.

We did incorporate a targeting moiety in our device, but although RVG-Lamp2b was reported to target exosomes to the brain[37], we did not find that it increased the delivery of therapeutic exosomes to this organ at least in the present context. However, we observed that cells equipped with the booster device were able to produce a sufficient amount of exosomes to rescue neurotoxicity and neuroinflammation associated with Parkinson's disease, without the requirement of directed targeting. This is significant, since many circulating exosomes are removed by the spleen and the liver[52].

If the repertoire of the available efficient targeting moieties and output molecules increases in the future, the engineered exosome producer cells available in vivo might open up new therapeutic opportunities for other intractable diseases as well, with the inherent ability of exosomes to reach every site in the body and even to cross the blood brain barrier[6]. Therapeutic designer exosomes may show improved safety compared to established nanoparticle-based or virus-based therapeutics, because exosomes are endogenous vesicles and the EXOtic devices are mostly human-derived.

Thus, the EXOtic devices developed in this study might serve as important tools to efficiently produce designer exosomes. Further, the implantable exosome producer cells engineered with the EXOtic devices could open up new therapeutic opportunities by enabling the delivery of therapeutic mRNA in situ in vivo.

## Methods

**EXOtic devices.** Comprehensive design and construction details for all expression vectors are provided in Supplementary Table 1. Key plasmids include the following. pDB60 encodes the exosome production booster ($P_{hCMV}$-STEAP3-IRES-SDC4-IRES-nadB fragment-pA). pSA465 encodes the RNA binding protein L7Ae conjugated to an exosomal marker CD63 ($P_{hCMV}$-CD63-L7Ae-pA). pSA462 encodes nluc bearing 1 repeat of C/$D_{box}$ which binds to L7Ae ($P_{hCMV}$-nluc-C/$D_{box}$-pA). pDB68 encodes Cx43 S368A, a constitutively active mutant of Cx43 ($P_{hCMV}$-Cx43 S368A-pA). pRVG-Lamp2b encodes engineered Lamp2b bearing a RVG peptide in its N-terminus with glycosylation signal (GNSTM), Addgene #71294 ($P_{hCMV}$-RVG-Lamp2b-pA). pDB129 encodes catalase bearing a C/$D_{box}$ in its 3′UTR ($P_{hCMV}$-Catalase-C/$D_{box}$-pA). The oligonucleotides used for plasmid cloning are summarized in Supplementary Table 2.

**Cell culture and transfection.** HEK-293T cells (DSMZ: ACC-635), human mesenchymal stem cells transgenic for the catalytic subunit of human telomerase (hMSC-TERT[53]), HeLa cells (ATCC: CCL-2), and Neuro2A cells (ATCC: CCL-131) were cultivated in DMEM (Invitrogen) supplemented with 10% (v/v) fetal bovine serum (FBS, Sigma-Aldrich) and 1% (v/v) penicillin/streptomycin solution (Sigma-Aldrich) at 37 °C in a humidified atmosphere containing 5% CO$_2$. Murine microglial BV2 cells (ICLC, ATL03001) were cultured in RPMI (Life Technologies), supplemented with 10% (v/v) fetal bovine serum (FBS, Sigma-Aldrich) and 1% (v/v) penicillin/streptomycin solution (Sigma-Aldrich) at 37 °C in a humidified atmosphere of 5% CO$_2$ in air. Chinese hamster ovary cells (CHO-K1, ATCC: CCL-61) were cultivated in ChoMaster HP1 medium (Cell Culture Technologies) supplemented with 2 mM glutamine and 1% (v/v) penicillin/streptomycin solution. For serial passage of these cells, 0.05% trypsin-EDTA (Gibco) was used. Patient-derived hMSCs[54] were cultured in DMEM/F-12 GlutaMAX (ThermoFisher Scientific) containing 10 % FBS, and 1% (v/v) penicillin/streptomycin solution (4 ng/mL of basic fibroblast growth factor (Invitrogen) were also added for propagation. This was not added for the mRNA transfer experiments.)

For transfection (HEK-293T, hMSC-TERT, HeLa), $1.25 \times 10^5$ cells (counted with a Casy® TTC Cell Counter) were seeded per well on a 24-well plate at 24 h before transfection and 50 μL of DNA-polyethyleneimine (PEI) mixture that was produced by incubating 2.5 μL PEI (PEI, 20000 MW, Polysciences; stock solution 1 mg/mL in dH$_2$O) with 500 ng of total DNA (see the section on reporter gene assays for details), vortexing for 1 sec and incubating at r.t. for 15 min. When necessary, transfection mix and cells used for transfection were scaled up to 12-well plates, 6-well plates, 10-cm dishes, or 15-cm dishes, and the amounts of the reagents were

changed accordingly. Before transfection, cell culture medium was exchanged to fresh DMEM medium containing 10% FBS. The cells were incubated with the transfection mixture for 8–16 h at 37 °C and 5% CO$_2$. Subsequently, the medium was exchanged again for fresh, pre-warmed medium, allowing for the expression of the gene of interest and production of engineered exosomes.

Electroporation (patient-derived hMSCs) was done with the Neon electroporation system (Thermofisher Scientific). About $3 \times 10^5$ cells were suspended in electroporation buffer and 2000 ng of plasmid to be transfected was added. Then, electroporation was done under the following conditions: 3 pulses of 20 ms at 1200 mV. After the electroporation, the cells were seeded in 6-well plates. 6 h later, the medium was changed to fresh, pre-warmed medium, allowing for the expression of the genes of interest and production of engineered exosomes.

**Exosome production assay with the reporter CD63-nluc.** The supernatant of exosome producer cells was collected 24-36 h after medium change following the transfection. After stepwise centrifugation to remove cells and cell debris from the sample (300x$g$ 5 min, 2000x$g$ 10 min, 10,000x$g$ for 30 min), the supernatant was assayed. The protocol for each figure is as follows.

Figure 1b: 125 ng of pDB30 ($P_{hCMV}$-CD63-nluc-pA) and potential exosome production booster (375 ng (with the ratio indicated in the figure in the case of using multiple components) in total; pEYFP-C1 was used as a control) per well were co-transfected into HEK-293T cells in a 24-well plate with PEI. At 16 h after transfection, medium was changed to fresh DMEM. At 24 h after the medium change, nluc in the supernatant was measured.

Figure 1d: 700 ng of pDB30 ($P_{hCMV}$-CD63-nluc-pA) and 1300 ng of pDB60 ($P_{hCMV}$-STEAP3-IRES-SDC4-IRES-nadB fragment-pA) per well were co-transfected into patient-derived hMSCs by electroporation in a 6-well plate. Six hours after transfection, the medium was changed to fresh DMEM/F12 containing 10% FBS. At 48 h after the medium change, nluc in the supernatant was assayed.

**mRNA transfer assay.** Supernatant of exosome producer cells was collected 24–48 h after medium change after transfection. This supernatant was transferred to target cells prepared in 48-well or 24-well plates (60–80% confluency) after centrifugation to remove cells and cell debris. After 24 h, the receiver cells were trypsinized, spun down, and suspended in PBS (25 μL/cells from each well of a 48-well plate). 7.5 μL of the suspension were pipetted into a 384-black well plate, and 7.5 μL quantification reagent (0.15 μL Nano-Glo® Luciferase Assay Substrate per 7.5 μL of Nano-Glo® Luciferase Assay Buffer) were added. After 5 min, luminescence measurements were performed at 25 °C on a Tecan Infinite® M200 PRO with an integration time of 1 s per well. The luminescence value was quantified as relative light unit (RLU). The specific procedure for each figure was as follows.

Figure 2b: 125 ng of pSA465 ($P_{hCMV}$-CD63-L7Ae-pA), 190 ng of pDB60 ($P_{hCMV}$-STEAP3-IRES-SDC4-IRES-nadB fragment-pA), 50 ng of pRVG-Lamp2b ($P_{hCMV}$-RVG-Lamp2b-pA), 125 ng of pSA462 ($P_{hCMV}$-nluc-C/$D_{box}$-pA) and 50 ng of pDB68 ($P_{hCMV}$-Cx43 S368A-pA) per well were co-transfected into HEK-293T cells in a 24-well plate with PEI (pEYFP-C1 was used to replace missing components). At 16 h after transfection, the medium was changed to fresh DMEM (650 μL). At 30 h after the medium change, the cell culture supernatant was harvested. The supernatant was centrifuged at 300x$g$ for 5 min and 500 μL of the supernatant were carefully transferred to a new tube. Again, the new tube was centrifuged at 2000x$g$ for 5 min and 400 μL of the supernatant was taken. Two hundred microliter of fresh DMEM was added, and the total 600 μL of the medium containing engineered exosomes was applied to receiver cells (per well of a 48-well plate, see below for preparation). After 24 h, the receiver cells were assayed for nluc activity as described above.

Figure 2c: HEK-293T cells in a 24-well plate were transfected with the following plasmids (per well). One hundred and twenty-five nanogram of pSA465 ($P_{hCMV}$-CD63-L7Ae-pA), 190 ng of pDB60 ($P_{hCMV}$-STEAP3-IRES-SDC4-IRES-nadB fragment-pA), 50 ng of pRVG-Lamp2b ($P_{hCMV}$-RVG-Lamp2b-pA), 50 ng of pDB68 ($P_{hCMV}$-Cx43 S368A-pA) and 125 ng of one of the following nluc-expressing plasmids. 0 repeat: pRK0 ($P_{hCMV}$-nluc-pA), 1 repeat: pSA462 ($P_{hCMV}$-nluc-C/$D_{box}$-pA), 3 repeats: pSA463 ($P_{hCMV}$-nluc-3xC/$D_{box}$-pA), 5 repeats: pSA464 ($P_{hCMV}$-nluc-5xC/$D_{box}$-pA). Then, the assay procedure was conducted as in Fig. 2b.

Figure 2d: For the "with device" condition, 300 ng of pSA465 ($P_{hCMV}$-CD63-L7Ae-pA), 600 ng of pDB60 ($P_{hCMV}$-STEAP3-IRES-SDC4-IRES-nadB fragment-pA), 120 ng of pRVG-Lamp2b ($P_{hCMV}$-RVG-Lamp2b-pA), 860 ng of pSA462 ($P_{hCMV}$-nluc-C/$D_{box}$-pA) and 120 ng of pDB68 ($P_{hCMV}$-Cx43 S368A-pA) per well were co-transfected into patient-derived hMSCs by electroporation in six-well plates. For the "no device" condition, 1120 ng of pEYFP-C1, 120 ng of pRVG-Lamp2b, and 860 ng of pSA462 per well were co-transfected into patient-derived hMSCs by electroporation in six-well plates. Six hours after transfection, the medium was changed to 2.5 mL of fresh DMEM/F12 containing 10% FBS. At 48 h after the medium change, the cell culture supernatant was harvested. The supernatant was centrifuged at 300x$g$ for 5 min and 2.2 mL of the supernatant were carefully transferred to a new tube. Again, the new tube was centrifuged at 2000x$g$ for 5 min and 2 mL of the supernatant were taken. One milliliter of fresh DMEM was added, and the total 500 μL of the medium containing the engineered exosome was applied to target cells/well of a 48-well plate (see below for preparation) (This protocol means twice the amount of exosome-containing supernatant was used for

this experiment compared to the other experiments of Fig. 2). After 24 h, the target cells were assayed for nluc activity as described above.

**Preparation of target cells for mRNA delivery.** HEK-293T cells were transfected with 400 ng of pDB63 (P$_{hCMV}$-CHRNA7-pA) and 1600 ng of EYFP-C1 (per well of a six-well plate). One day after transfection, cells were trypsinized, centrifuged, and re-suspended in double the amount of fresh DMEM (for example, re-suspend cells from a well of a six-well plate in 4 mL fresh DMEM, which is double the amount of 2 mL used for cell culture in each well of a six-well plate). Then, the cells were seeded in a 48-well plate (200 μL of suspension/well). Twenty-four hours after seeding, the cells were used for the experiment.

**Nanoparticle tracking analysis.** HEK-293T cells were transfected with the following plasmids (amounts of plasmids are per well of a six-well plate; six-wells were prepared for each condition). +production booster condition: 500 ng of pDB30 (P$_{hCMV}$-CD63-nluc-pA) and 1500 ng of pDB60 (P$_{hCMV}$-STEAP3-IRES-SDC4-NadB-pA). −production booster condition: pDB30 (P$_{hCMV}$-CD63-nluc-pA) and 1500 ng of pEYFP-C1 (P$_{hCMV}$-EYFP–pA). After overnight transfection, the medium was changed to 2.5 mL/well DMEM containing 10% exosome-depleted FBS (cat. no. A25904DG, further filtered with a 200 nm filter). After 48 h, cell culture supernatants were harvested (supernatant from two wells were combined: 5 mL/sample, triplicates for each condition). The cell culture supernatant was centrifuged at 300x*g* for 10 min to remove cells and the supernatant was carefully collected. Next, the supernatant was centrifuged at 20,000x*g* for 30 min to remove cell debris and the resulting supernatant was carefully collected. Exosomes were precipitated and purified by size exclusion chromatography with the Exo-spin Kit (Cell Guidance Systems) according to the manufacturer's instructions. The eluate with 200 μL of filtered DPBS was used for NTA with a Nanosight LM10 (Nano-Sight Ltd.), followed by evaluation using the Nanoparticle Tracking Analysis (NTA) software with the help of Verena Christen (FHNW, Basel, Switzerland). Conditions were as follows: camera type, CCD; detection threshold, 5-Multi; recording for 1800 frames at 30 frames/s.

**Neurotoxicity rescue in 6-OHDA Parkinson's disease model.** For the in vitro rescue of experimental Parkinson's disease, HEK-293T cells (exosome producer cells) were transfected with the following plasmids (amounts of plasmid are per well of a 12-well plate). Catalase delivery: 150 ng of pSA465 (P$_{hCMV}$-CD63-L7Ae-pA), 300 ng of pDB60 (P$_{hCMV}$-STEAP3-IRES-SDC4-IRES-nadB-pA), 60 ng of pRVG-Lamp2b (P$_{hCMV}$-RVG-Lamp2b-pA), 430 ng of pDB129 (P$_{hCMV}$-Catalase-C/D$_{box}$-pA) and 60 ng of pDB68 (P$_{hCMV}$-Cx43 S368A-pA). Mock: pcDNA3.1(+) 1000 ng. Three wells of a 12-well plate were prepared for each condition. After overnight transfection, the medium was changed to fresh DMEM (1.2 mL/well). Thirty hours later, the cell culture supernatant was harvested (total 3.6 mL/condition) and centrifuged at 2000x*g* for 5 min. The supernatant (3.2 mL/condition), was carefully collected, and 2 mL of fresh DMEM was added. This conditioned medium was added to Neuro2A cells seeded in a 96-well plate (200 μL/well) in triplicate (1.88 × 10$^4$ cells of Neuro2A cells were seeded per well of a 96-well plate one day before this medium transfer). After 24 h, 0–1500 μM 6-hydroxydopamine (6-OHDA) solution in DMEM was added to the cells (final: 0–300 μM). At 3 h after addition of 6-OHDA, the medium was changed to 200 μL of fresh DMEM. Twenty-four hours later, the medium was replaced with fresh DMEM containing 10% CCK-8 assay solution, and CCK-8 assay was performed according to the manufacturer's protocol. The measurement of absorbance was performed at 450 nm with an EnVision 2104 plate reader.

**Nanoluc mRNA delivery.** HEK-293T cells in a 15-cm dish were transfected with the following plasmids (2 dishes were prepared per condition; plasmid amount is per 15 cm dish). With device: 2750 ng of pSA465 (P$_{hCMV}$-CD63-L7Ae-pA), 5500 ng of pDB60 (P$_{hCMV}$-STEAP3-IRES-SDC4-IRES-nadB fragment-pA), 1100 ng of pRVG-Lamp2b (P$_{hCMV}$-RVG-Lamp2b-pA), 9750 ng of pSA462 (P$_{hCMV}$-nluc-C/D$_{box}$-pA), and 1100 ng of pDB68 (P$_{hCMV}$-Cx43 S368A-pA). Without device: 10450 ng of pEYFP-C1 (P$_{hCMV}$-EYFP-pA), 9750 ng of pSA462 (P$_{hCMV}$-nluc-C/D$_{box}$-pA). After overnight transfection, the cells were trypsinized (two dishes/group were combined), spun down, and re-suspended in 1.8 mL DMEM without FBS. The cell suspension was put on ice and mixed with 1.8 mL of ice-cold Matrigel (Corning). Four hundred microliter of the cell/Matrigel mixture was injected s.c. into a C57BL/6 J mice (8 weeks of age, female, total 8 mice). Forty-eight hours later, the mice were killed by overdose anesthesia and whole-body perfusion was performed with PBS. The brain, spleen, liver of each mouse were removed and put into 500 μL (for brain and liver) or 300 μL (for spleen) of DPBS containing 1% protease inhibitor cocktail (Sigma). The tissue was mechanically homogenized and centrifuged at 9000x*g* for 20 min at 4 °C. The supernatant was used for nluc assays (7.5 μL of the supernatant +7.5 μL of assay solution) on a Tecan Infinite M200 Pro plate reader.

**Catalase mRNA delivery and attenuation of neuroinflammation.** Common procedure for Fig. 3e, f:
HEK-293T cells in a 15-cm dish were transfected with the following plasmids (2 dishes were prepared per condition; plasmid amount is per 15-cm dish): 2750 ng of

pSA465 (P$_{hCMV}$-CD63-L7Ae-pA), 5500 ng of pDB60 (P$_{hCMV}$-STEAP3-IRES-SDC4-IRES-nadB fragment-pA), 1100 ng of pRVG-Lamp2b (P$_{hCMV}$-RVG-Lamp2b-pA), 9750 ng of pDB129 (P$_{hCMV}$-Catalase-C/Dbox-pA). (For the (−) condition, corresponding plasmids were replaced with pEYFP-C1.) After overnight transfection, the cells were trypsinized (4 × 10$^7$ cells/group), spun down, and re-suspended in 2 mL DMEM without FBS. The cell suspension was put on ice, and mixed with 2 mL of ice-cold Matrigel (Corning). Three hundred and fifty microliter of the cell/Matrigel mixture was injected s.c. into the C57BL/6 J mice (8 weeks of age, female) (approx. $3.5 \times 10^6$ cells/mouse). One day later, the mice were intracerebrally injected with 500 nL (bilaterally, per side) of a 5 mg/mL 6-OHDA solution in 0.9% (w/v) NaCl with 0.02 % (w/v) ascorbic acid (flow rate of 0.1 μL/min into the striatum (AP: + 0.5; L: −2.0 and DV: −3.0 mm)). At 6 days after 6-OHDA injection, the mice were deeply anesthetized with an i.p. injection of pentobarbital (300 mg/kg) and perfused intra-cardially with cold PBS for Fig. 3e or cold PBS and 4% paraformaldehyde (PFA) for Fig. 3f.

Figure 3e specific protocol after the above process:
The brain of each mouse was removed and placed in 400 μL of TRIzol Reagent (ThermoFisher Scientific). The tissue was mechanically homogenized and frozen with dry ice until the RNA extraction process. For RNA extraction, the homogenate was thawed on ice. After a centrifugation step (5 min at 12,000x*g* and 4 °C), 50 μL of the homogenate was diluted with 450 μL of the TRIzol reagent and RNA was extracted following the reagent manufacturer's protocol. Then, cDNA was prepared using High Capacity cDNA Reverse Transcription Kits (Applied Biosystems) with RiboLock RNase Inhibitor (Thermofisher Scientific). The amounts of GFAP, Iba1, TNFα, CD11b, and GAPDH mRNA were measured by qPCR (Mastercycler ep Realplex Real-time PCR) using the following qPCR kits; Taqman qPCR probes were used for GAPDH and TNFα, and SYBR Green Master mix (Applied Biosystems) was used for the other genes (the melting curve was confirmed for all genes). Primers: for GFAP: Fw: ATCGAGATCGCCACCTACAG, Rev.: CTCACATCACCACGTCCTTG[55, 56], IbaI: Fw: CAGACTGCCAGCCTAAGACA Rev: AGGAATTGCTTGTTGATCCC[57], TNFα Taqman qPCR probe Mm00443258_m1, GAPDH: Taqman qPCR probe Mm99999915_g1. All mRNA expression levels were normalized by GAPDH mRNA expression. Further, each mRNA expression level was normalized to that of the control group (no 6-OHDA, no implant).

Figure 3f specific protocol: The brains were extracted and left overnight in 4% PFA at 4 °C. Twenty-four hours after perfusion, the brains were transferred to 30% sucrose and kept at 4 °C until saturated. Sixty micrometer slices containing the dorsal striatum or the midbrain were obtained in a cryostat (Reichert Jung; Cryocut 1800) and processed for immunohistochemistry. Free-floating slices were washed with 0.1% Tween in TBS (TBST), permeabilized in 10% Triton in TBST and then washed again in TBST. Following a 2-h blocking incubation in 5% bovine serum albumin in TBS at room temperature, the slices were incubated for 24 h at 4 °C with a polyclonal antibody against tyrosine hydroxylase raised in rabbit (Sigma T8700-1VL). After the incubation with the primary antibody, the slices were again washed in TBST and then incubated for 2 h at room temperature with a secondary anti-rabbit antibody raised in goat (Alexa Fluor 555; A32732). The slices were finally washed in TBST and mounted on glass slides for confocal microscopy.

Figure 3g: The same exosome producer cells as used for Fig. 3e were s.c. injected into C57BL/6 J mice (8 weeks of age, female, total 8 mice, cells from two 15-cm dishes/mouse). 2 days later, the mice were i.p. injected with 150 μL of DPBS containing 0.3 mg/kg LPS-EB (from E. coli O111:B4, InvivoGen) (for the (−) condition, only DPBS was injected). Four hours later, the mice were killed with an overdose of anesthesia. The brain homogenate was prepared with the same method as described for Fig. 3e, and GFAP mRNA expression level was monitored with the same method.

**Ethical approval.** Experiments involving animals were carried out in accordance with the directive of the European Union by Ghislaine Charpin-El Hamri and Marie Daoud-El Baba (No. 69266309; project No. DR2013–01 (v2)) at the Institut Universitaire de Technologie, UCB Lyon 1, F-69622 Villeurbanne Cedex, France and in accordance with the Institutional Animal care of the University of Basel with authorization of the Cantonal Veterinary office by Kelly R. Tan and Giorgi Rizzi.

**Data availability.** All data and materials are available upon request.

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

## Acknowledgements

The authors thank Dr. O. Wiklander (Karolinska Institute, Sweden) and Verena Christen (FHNW, Basel, Switzerland) for the help with the NTA assay, Dr. Bernard Schneider for providing the macroencapsulation device, and Addgene construct suppliers for providing plasmids (see SI for details). This work was supported by the European Research Council (ERC) advanced grant (ProNet, no. 321381) and in part by the National Centre of Competence in Research (NCCR) for Molecular Systems Engineering. In vivo

experiments conducted by K.T. and G.R. were supported by the Swiss National Science Foundation (SNSF, PP00P3-1150683). R.K. was supported by a postdoctoral fellowship from the Human Frontier Science Program (HFSP, LT000094/2014-L).

## Author contributions

R.K., D.B. and M.F. designed the project, analyzed results and wrote the manuscript. R.K and D.B. conducted the in vitro experimental work. R.K., D.B., G.C.H., M.E.B, G.R., and K.T. planned and performed the animal experiments. P.S. helped conduct experiments involving stem cells and S.A. constructed specific plasmids.

## Additional information

**Competing interests:** The authors declare no competing interests.

