## [Peer Review File(PDF 300 kb) · Nature Communications]

Reviewers' comments:

Reviewer #1 (Remarks to the Author):

Summary:

The authors engineered an exosome-producing genetic system (EXOtic) that, when transiently transfected into mammalian cells, is capable of producing exosomes that can deliver therapeutic mRNA to target cells in vitro and in vivo. To accomplish this, they designed a fluorescent based screen for exosome production where a fluorophore, nluc, was conjugated to an exosome marker, CD63. Using their system, the authors enhance the production of exosomes by as much as 40-fold through tri-cistronic expression of key genes that regulate exosomes. In addition to mechanistic work in-vitro, they further demonstrate the potential of this technology with an in-vivo model of Parkinson's disease. The figure illustrations are of noteworthy quality.

Major comments to the authors:

1. The impact of using exosomes as a therapeutic for treating Parkinson's disease would be greater if the authors discussed in their introduction that exosomes readily cross the blood-brain barrier, rather than waiting for the discussion.
2. More clarification for the rationale of moving between cell lines would improve the flow of the manuscript. Why did the authors decide not to move forward with hMSCs in the in vitro and in vivo models if "hMSCs are used as a source for therapeutic exosomes"? Additionally, their data supports good expression of exosomes using their booster. This is curious.
3. In figure 1b, it is difficult to interpret the data because of the choice of x-axis labeling. Instead of labeling by letter the authors might want to consider labeling the axis: Control, NadB, SDC4, etc.
4. Regarding the data in figure 1c, the literature size for exosomes has been reported as 30-100nm with larger particles described as microvesicles. The authors report a distribution of larger particles in figure 1c. The authors should provide commentary on how their distribution relates to the literature, and whether it is possible that the particles could be other extracellular vesicles or apoptotic bodies. Perhaps an ELISA, Western blot, or flow cytometry could be used to determine if the cell culture supernatant has an increase in proteins associated with exosomes specifically?
5. Regarding the data in figure 2a, the schematic is nicely done but the order is possibly missed labeled. Production and secretion would presumably occur after RNA packaging. This is confusing - possibly separate "production & secretion"? This also relates to the first paragraph - it would read better if the authors identify the 4 processes that require optimization in the order in which they occur in cells. It is confusing that "secretion" is listed first followed by packaging of mRNAs into exon. Also, the stars indicating luciferase should be defined in the legend.
6. In figure 3e, the catalase is actually catalase mRNA, and should be labeled appropriately.
7. The claim in the discussion that mRNA delivery can attenuate neurotoxicity is either not fully explained or not fully supported by reducing the levels of one mRNA.
8. One of the novelties of the EXOtic device is targeted delivery. However, in the in vivo experiments, the authors only assess mRNA expression in the brain of each mouse. Were other tissues analyzed for effects of catalase mRNA delivery? It is hard to accept targeted delivery without examination of mRNA levels in other organs.

Minor comments to the authors:

1. The abstract would read better if the first sentence was a description of exosomes rather than having their definition in parentheses.

2. The second sentence in the abstract would read better if the parentheses were removed and replaced by commas.
3. This manuscript reads well, but the authors might want to consider moving away from "we did..." This would also help with the limited space allowed for this particular journal.
4. Regarding figure 2c, the denoted (no C/Dbox) is hard to read. The authors should change font size or add an asterisks to the + sign to direct readers to the explanation in the legend.
5. Since the novel aspect of this work is the exosomes, the authors may want to consider changing the title so that the focus is on exosomes rather than the cells.
6. Consider discussing the safety of the EXOtic devices in the discussion.
7. Consider referencing figure 3c after the 1st sentence in the 1st paragraph on page 3
8. The author's claims that catalyse mRNA delivery via EXOtic devices attenuates neuroinflammation and neurotoxicity would be supported better by a description of the basic biology of Parkinson's disease and viability of exosomes across the blood brain barrier.

Reviewer #2 (Remarks to the Author):

The manuscript by Kojima et al. explores the use of a number of 'devices' for the increased production, packaging and targeting of exosomes for therapeutic use. The authors have used a number of constructs in their studies to produce cells that produce exosomes that have been used in both in vitro and in vivo assays. The manuscript is complex given the wide range of pathways investigated to engineer exosomes, but the authors have provided detailed explanations and diagrams that assist the reader. The exosomes produced contain RNA that has been used to deliver RNA encoding the enzyme catalyse that the authors has tested in what they refer to as Parkinson's disease models.

Questions:

- (1) The authors describe the use of a exosome production booster expressing three different constructs, it would be worthwhile to have a short explanation in the manuscript for how each of the proteins are thought to function to increase the production of exosomes.
- (2) In figure 1b the tricistronic vector was shown to significantly increase the production of exosomes compared to expressing each of the constructs together. Western blotting to identify the expression of these proteins is required to show the level of each protein that adds to the increase in exosome production. Currently it is not clear which proteins are functioning in exosome production given that the data already suggests that different levels of protein expression can have such a large affect.
- (3) Further characterisation of the luminescence assay developed for the quantification of exosomes is required. The nanoluc system is very sensitive and stable, as such the authors need to show that the CD63-nanoluc protein is not being exported from the cell independent of exosomes. This could be achieved by comparing the signal of the cell supernatant both before and after removal of exosomes. This is critical given the importance of this assay in the manuscript.
- (4) The authors use the designer exosomes to rescue two assays referred to as being models of Parkinson's disease. A 6-OHDA assay is used in vitro to show a positive affect for the catalase packaged exosomes. It is therefore unusual that in the next in vivo assay the authors changed from using 6-OHDA (a standard Parkinson's disease in vivo assay) to using LPS, which has limited application to PD. What was the rationale for not using an in vivo 6-OHDA assay?
- (5) The in vivo studies use cells delivered in matrigel, as such the cells have the potential to

migrate from the injection site. Have the authors considered this possibility?

(6) In figure 3e RNA levels of GFAP are used to show protection from neuroinflammation, however, is a two-fold decrease in GFAP RNA levels enough to result in a reduction of neuroinflammation—particularly for PD? It is unclear to me if a single assay measuring the RNA levels of a single gene can represent such a complex process. RNA levels of other genes involved in neuroinflammation or better the protein levels or astrocyte levels (using immunohistochemistry) would add support to these findings.

Minor points:

In the abstract the authors state that exosomes range from 30-200nm. This range is very large and outside the normal description of between 50-150nm (or even less).

For a number of figures no statistics are shown (for example figure 2b-d). Although they may be very significant changes the statistical analysis should be performed.

NCOMMS-17-1878A - Response to the reviewers' comments

Reviewer 1

“The authors engineered an exosome-producing genetic system (EXOtic) that, when transiently transfected into mammalian cells, is capable of producing exosomes that can deliver therapeutic mRNA to target cells in vitro and in vivo. To accomplish this, they designed a fluorescent based screen for exosome production where a fluorophore, nluc, was conjugated to an exosome marker, CD63. Using their system, the authors enhance the production of exosomes by as much as 40-fold through tri-cistronic expression of key genes that regulate exosomes. In addition to mechanistic work in-vitro, they further demonstrate the potential of this technology with an in-vivo model of Parkinson's disease. The figure illustrations are of noteworthy quality.”

Major comments to the authors:

1. “The impact of using exosomes as a therapeutic for treating Parkinson's disease would be greater if the authors discussed in their introduction that exosomes readily cross the blood-brain barrier, rather than waiting for the discussion.”

As requested, we now mention that exosomes can cross the blood-brain barrier in the introduction.

2. “More clarification for the rationale of moving between cell lines would improve the flow of the manuscript. Why did the authors decide not to move forward with hMSCs in the in vitro and in vivo models if “hMSCs are used as a source for therapeutic exosomes”? Additionally, their data supports good expression of exosomes using their booster. This is curious.

We chose HEK-293T cells for all in vivo applications because of their higher exosome production capacity, increased transfection rate and cultivation robustness. HEK-293T cells are currently the preferred parental cell line for designer cell-based proof-of-concept studies in vivo (e.g., Kemmer et al., 2010, Nat. Biotechnol. 28:355; Rössger et al., 2013, Nat. Commun. 4:2825; Ye et al., 2017, Nat. Biomed. Eng. 1:5; Xie et al., 2016, Science 354:11296). We added this information in the revised manuscript.

3. “In figure 1b, it is difficult to interpret the data because of the choice of x-axis labeling. Instead of labeling by letter the authors might want to consider labeling the axis: Control, NadB, SDC4, etc.”

We have re-labeled the x-axis as suggested.

4. “Regarding the data in figure 1c, the literature size for exosomes has been reported as 30-100nm with larger particles described as microvesicles. The authors report a distribution of larger particles in figure 1c. The authors should provide commentary on how their distribution relates to the literature, and whether it is possible that the particles could be other extracellular vesicles or apoptotic bodies. Perhaps an ELISA, Western blot, or flow cytometry could be used to determine if the cell culture supernatant has an increase in proteins associated with exosomes specifically?”

We have repeated the Nanoparticle Tracking Analysis (NTA) experiment using a commercial exosome purification kit (Exo-Spin kit; Cell Guidance Systems Ltd., Cambridge, UK) that is based on size exclusion chromatography and yields purification standards comparable with ultracentrifugation (Santangelo et al., 2016, Cell Reports 17:799; Lane et al., 2015, Scientific Reports 5:7639). This enabled us to collect smaller-sized vesicles that are within the exosome range reported in the literature (e.g., Peinado et al, Nat. Rev. Cancer, 2017, 17, 302. and Haraszti et al. J. Extracell. Vesicles 2016, 5, 32570).

We have further confirmed (i) that our device substantially increases the production of exosomes without changing their size distribution (new Fig. 1c, new Fig. S3), (ii) that it boosts exosome secretion, as indicated by ELISA-based profiling of exosome-specific markers such as CD9 and TSG101, as well as the cytosolic negative-control marker HSP90B1 (new Fig. S2d-f), and (iii) that booster capacity is retained when the alternative CD9-nluc reporter is used (new Fig. S2c). In addition, we confirmed (iv) that the luciferase signal was indeed associated with exosomes (new Fig. S2a,b). This set of new control experiments corroborates that exosome production is indeed enhanced by the exosome production booster device.

5. “Regarding the data in figure 2a, the schematic is nicely done but the order is possibly missed labeled. Production and secretion would presumably occur after RNA packaging. This is confusing - possibly separate “production & secretion”? This also relates to the first paragraph – it would read better if the authors identify the 4 processes that require optimization in the order in which they occur in cells. It is confusing that “secretion” is listed first followed by packaging of mRNAs into exon. Also, the stars indicating luciferase should be defined in the legend.”

Thank you. We have revised Figure 2a and the corresponding main text as suggested.

6. “In figure 3e, the catalase is actually catalase mRNA, and should be labeled appropriately.”

Re-labeled as requested.

7. “The claim in the discussion that mRNA delivery can attenuate neurotoxicity is either not fully explained or not fully supported by reducing the levels of one mRNA.”

We have done extensive additional in vivo experiments using another in vivo Parkinson’s disease model based on intracerebral injection of 6-OHDA. The 6-OHDA mouse model showed rescue of neuronal cell death in the striatum (new Fig. 3f). We have also confirmed downregulation of several neuroinflammation-specific markers (GFAP, Iba1, TNF α , and CD11b) after exosome-based catalase mRNA delivery in vivo (new Fig. 3e).

For consistency with the in vivo experiment using systemic LPS injection (new Fig. 3g; the same as old Fig. 3e), we also show the protective effect of exosome-based delivery of catalase mRNA in vitro using LPS as a toxin (new Fig. S7).

Overall, we now report consistent protective/rescuing effects of exosome-based delivery of catalase mRNA cargo for two major in vitro and in vivo neurotoxic models of Parkinson’s disease: the LPS model and the 6-OHDA model.

8. “One of the novelties of the EXOtic device is targeted delivery. However, in the in vivo experiments, the authors only assess mRNA expression in the brain of each mouse. Were other tissues analyzed for effects of catalase mRNA delivery? It is hard to accept targeted delivery without examination of mRNA levels in other organs.”

We have done an additional experiment to profile the biodistribution of the exosome-based luminescence signal in the liver and the spleen, which typically clear nano-sized particles such as exosomes from the bloodstream (Barile and Vasally, 2017, Pharmacol. Ther., new Fig. S9).

As regards targeting: (i) We did not intend to pioneer targeting technology, but just used an established targeting strategy based on RVG-Lamp2b (Alvarez-Erviti et al. 2011, Nat. Biotechnol. 29:341; Hung and Leonard 2015, J. Biol. Chem. 290:8166.). (ii) In fact, however, unlike in vitro, we found that RVG-Lamp2b’s targeting efficacy was poor, as has already been noted elsewhere (Wiklander et al., 2015, J. Extracell. Vesicles 4:26316), and it did not increase the delivery of therapeutic exosomes to the brain. (iii) We added a comment in the main text, pointing out that many exosomes are cleared by the liver and the spleen (Wiklander et al., 2015, J. Extracell. Vesicles

4:26316; Barile and Vasally, 2017, *Pharmacol. Ther.* 174:63; McKelvey et al., 2015, *J. Circ. Biomark.* 4:7).

Overall, we think that the issue of targeting has become moot, at least in the present context, because we have demonstrated that the production booster device produces sufficient cargo-containing exosomes to provide a protective/rescue effect in two major in vivo models of Parkinson's disease (pls. see point 7 above). We have discussed the above-mentioned points in the revised version of the manuscript and pointed out that a variety of targeting strategies may be developed in the future for other exosome-based therapeutic strategies.

Minor comments to the authors

1. "The abstract would read better if the first sentence was a description of exosomes rather than having their definition in parentheses."

We have changed the sentence as requested.

2. "The second sentence in the abstract would read better if the parentheses were removed and replaced by commas."

We have changed the sentence as requested.

3. "This manuscript reads well, but the authors might want to consider moving away from "we did..." This would also help with the limited space allowed for this particular journal."

We have modified the wording accordingly.

4. "Regarding figure 2c, the denoted (no C/Dbox) is hard to read. The authors should change font size or add an asterisks to the + sign to direct readers to the explanation in the legend."

This point has been addressed as suggested.

5. "Since the novel aspect of this work is the exosomes, the authors may want to consider changing the title so that the focus is on exosomes rather than the cells."

We have changed the title.

6. "Consider discussing the safety of the EXOtic devices in the discussion."

We have included this point in the revised discussion.

7. "Consider referencing figure 3c after the 1st sentence in the 1st paragraph on page 3."

Done as requested.

8. "The author's claims that catalase mRNA delivery via EXOtic devices attenuates neuroinflammation and neurotoxicity would be supported better by a description of the basic biology of Parkinson's disease and viability of exosomes across the blood brain barrier."

We edited the introduction to provide the requested information.

Reviewer 2

“The manuscript by Kojima et al. explores the use of a number of ‘devices’ for the increased production, packaging and targeting of exosomes for therapeutic use. The authors have used a number of constructs in their studies to produce cells that produce exosomes that have been used in both in vitro and in vivo assays. The manuscript is complex given the wide range of pathways investigated to engineer exosomes, but the authors have provided detailed explanations and diagrams that assist the reader. The exosomes produced contain RNA that has been used to deliver RNA encoding the enzyme catalase that the authors has tested in what they refer to as Parkinson’s disease models.”

1. “The authors describe the use of a exosome production booster expressing three different constructs, it would be worthwhile to have a short explanation in the manuscript for how each of the proteins are thought to function to increase the production of exosomes.”

We have added known activities of each component of the exosome production booster and included appropriate references.

2. “In figure 1b the tricistronic vector was shown to significantly increase the production of exosomes compared to expressing each of the constructs together. Western blotting to identify the expression of these proteins is required to show the level of each protein that adds to the increase in exosome production. Currently it is not clear which proteins are functioning in exosome production given that the data already suggests that different levels of protein expression can have such a large affect.”

Tricistronic vectors fix the expression level of each protein and ensure that every transfected cell receives all three booster components (Fussenegger et al., 1998, Nat. Biotechnol. 16:468). This is confirmed by direct comparison of booster performance between transfection of the tricistronic vector and co-transfection of three individual plasmids (Fig. 1b).

Comparative quantification of individual proteins produced from a multicistronic vector by Western blotting is challenging because each antibody has a target-specific affinity and tagging the proteins may bias protein production, stability and function (Kimple et al., 2013, Curr. Protoc. Protein Sci. 73:9.9.; Waugh et al., 2005, Trends Biotechnol. 23:316). Additionally, there is no commercial antibody available to quantify Nad B fragment.

Therefore, we have chosen only to discuss and clarify the aforementioned points in the revised version of the manuscript.

3. “Further characterization of the luminescence assay developed for the quantification of exosomes is required. The nanoluc system is very sensitive and stable, as such the authors need to show that the CD63-nanoluc protein is not being exported from the cell independent of exosomes. This could be achieved by comparing the signal of the cell supernatant both before and after removal of exosomes. This is critical given the importance of this assay in the manuscript.”

We now provide a comparative analysis of the CD63-nluc-based luminescence before and after precipitation of exosomes to confirm that the luminescence signal was indeed associated with exosome components (new Fig. S2a,b). Inspired by the reviewer’s point, we have further (i) confirmed that depletion of the exosome fraction also substantially decreased mRNA-cargo delivery (new Fig. S5), (ii) quantified the performance of the exosome production booster using a different reporter construct (CD9-nluc, new Fig. S2c) and (iii) confirmed the identity of exosomes by ELISA (new Fig. S2d-f).

4. “The authors use the designer exosomes to rescue two assays referred to as being models of Parkinson’s disease. A 6-OHDA assay is used in vitro to show a positive affect for the catalase packaged exosomes. It is therefore unusual that in the next in vivo assay the authors changed from using 6-OHDA (a standard Parkinson’s disease in vivo assay) to using LPS, which has limited application to PD. What was the rationale for not using an in vivo 6-OHDA assay?”

For the original submission, we chose systemic LPS injection as an in vivo Parkinson's disease model, due to restrictions imposed by our experimental ethical approval.

In collaboration with colleagues having the appropriate license and know-how, we have now conducted extensive additional in vivo experiments using another in vivo Parkinson's disease model based on intracerebral injection of 6-OHDA. These experiments have shown rescue of neuronal cell death in the striatum (new Fig. 3f) as well as downregulation of several neuroinflammation-specific markers (GFAP, Ibal, TNF α , and CD11b) after exosome-based catalase mRNA delivery (new Fig. 3e).

For consistency with the LPS-based in vivo model, we have also confirmed the protective effect of exosome-based delivery of catalase mRNA in vitro using LPS as a toxin (new Fig. S7).

Overall, we now report consistent protective/rescuing effects of exosome-based delivery of catalase mRNA in two major in vitro and in vivo neurotoxic models of Parkinson's disease: the LPS model and the 6-OHDA model (please also see our response to comment 7 of reviewer 1).

5. "The in vivo studies use cells delivered in matrigel, as such the cells have the potential to migrate from the injection site. Have the authors considered this possibility?"

We conducted additional control experiments to exclude this possibility. Implantation of exosome-producing cells inside a macroencapsulation device with 0.45 μ m pore-size (Lathulière et al., 2014 Biomaterials 35:779), enabling diffusion of exosomes while retaining producer cells, showed similar CD63-nluc-based luminescence in the blood and mRNA cargo delivery into the brain compared to producer cells implanted with Matrigel (new Fig. S9).

6. "In figure 3e RNA levels of GFAP are used to show protection from neuroinflammation, however, is a two-fold decrease in GFAP RNA levels enough to result in a reduction of neuroinflammation-particularly for PD? It is unclear to me if a single assay measuring the RNA levels of a single gene can represent such a complex process. RNA levels of other genes involved in neuroinflammation or better the protein levels or astrocyte levels (using immunohistochemistry) would add support to these findings."

In the new 6-OHDA-based in vivo experiment, we confirmed attenuation of neuroinflammation using multiple neuroinflammatory markers ((GFAP, Ibal, TNF α , and CD11b; new Fig. 3e) as well as rescue of neurotoxicity in the striatum (new Fig. 3f). These results firmly support the therapeutic effect of the designer exosomes (please also see our responses to point 4 of reviewer 2 and point 7 of reviewer 1).

Although the absolute fold changes of the neuroinflammation markers appear to be moderate, it should be noted that we extracted the mRNA from the whole brain, whereas 6-OHDA injection usually causes only localized neuroinflammation. We have included this point in the revised version of the manuscript.

Minor points

1. "In the abstract the authors state that exosomes range from 30-200nm. This range is very large and outside the normal description of between 50-150nm (or even less)."

We have revised the statement of exosome size in the abstract and repeated the Nanoparticle Tracking Analysis (NTA) experiment using a commercial exosome purification kit (Exo-Spin kit; Cell Guidance Systems Ltd., Cambridge, UK) to collect appropriately sized vesicles (new Fig. 1c, S3). (See also our response to comment 4 of reviewer 1).

2. "For a number of figures no statistics are shown (for example figure 2b-d). Although they may be very significant changes the statistical analysis should be performed."

We have now done statistical analyses for all results in the main test and the supporting information.

REVIEWERS' COMMENTS:

Reviewer #1 (Remarks to the Author):

The authors have done an exceptional job with their revisions. They have addressed all of this reviewer's concerns and comments. When extensive changes/edits are made to a manuscript during the review process, there can be problems with putting the manuscript back together. The manuscript reads well and the results are compelling. The authors have clearly taken the time to address all comments and write an excellent manuscript. Kudos to the authors for a job well done.

Reviewer #2 (Remarks to the Author):

The manuscript by Kojima et al has been substantially revised and I believe the authors have answered my main queries. Importantly, the authors have taken the time to include the 6-OHDA model in vivo (I understand the hurdles required for ethics in such a model) as well as to improve their data showing that the signal is indeed coming from the exosome fraction rather than the supernatant.

Minor points:

In line 123 the sentence 'Since exosomes can cross the blood-brain barrier' requires a reference. There are a couple of papers that I believe show this conclusively (preferably not using a dye labelling system but delivery of either RNA or a Cre based system).

Line 224 the sentence '...Parkinson's disease, the issue of targeting is moot, at least in the present context' should be changed. I agree with the authors that there is substantial evidence to suggest that RGV-Lamp2b does not target the brain specifically, however saying the issue of targeting is moot I do not feel is correct. Rather a rewording such as 'However, we observed that cells equipped with the booster device were able to produce a sufficient amount of exosomes to rescue neurotoxicity and neuroinflammation associated with Parkinson's disease, without the requirement of directed targeting. This is significant, given many circulating exosomes are removed by the spleen and the liver'.

In Figure 3F the abbreviation DStr should be explained.

NCOMMS-17-1878A - Response to the remaining reviewers' comments

Reviewer 1

Reviewer #1 (Remarks to the Author):

The authors have done an exceptional job with their revisions. They have addressed all of this reviewer's concerns and comments. When extensive changes/edits are made to a manuscript during the review process, there can be problems with putting the manuscript back together. The manuscript reads well and the results are compelling. The authors have clearly taken the time to address all comments and write an excellent manuscript. Kudos to the authors for a job well done.

We appreciate the reviewer's favorable comment on our revised manuscript.

Reviewer #2 (Remarks to the Author):

The manuscript by Kojima et al has been substantially revised and I believe the authors have answered my main queries. Importantly, the authors have taken the time to include the 6-OHDA model in vivo (I understand the hurdles required for ethics in such a model) as well as to improve their data showing that the signal is indeed coming from the exosome fraction rather than the supernatant.

We again appreciate the reviewer's favorable comment on our revision.

Minor points:

In line 123 the sentence 'Since exosomes can cross the blood-brain barrier' requires a reference. There are a couple of papers that I believe show this conclusively (preferably not using a dye labelling system but delivery of either RNA or a Cre based system).

We have added appropriate references in the corresponding part.

Line 224 the sentence '...Parkinson's disease, the issue of targeting is moot, at least in the present context' should be changed. I agree with the authors that there is substantial evidence to suggest that RGV-Lamp2b does not target the brain specifically, however saying the issue of targeting is moot I do not feel is correct. Rather a rewording such as 'However, we observed that cells equipped with the booster device were able to produce a sufficient amount of exosomes to rescue neurotoxicity and neuroinflammation associated with Parkinson's disease, without the requirement of directed targeting. This is significant, given many circulating exosomes are removed by the spleen and the liver'.

We have revised the corresponding part as suggested.

In Figure 3F the abbreviation DStr should be explained.

We explained the abbreviation of DStr in the figure legend.